# Performance of Feature-Based Techniques for Automatic Digital Modulation Recognition and Classification—A Review

**Dhamyaa H. Al-Nuaimi** [1,2], **Ivan A. Hashim** [3], **Intan S. Zainal Abidin** [1], **Laith B. Salman** [2] **and Nor Ashidi Mat Isa** [1,*]

[1] School of Electrical & Electronic Engineering, Engineering Campus, Universiti Sains Malaysia, Nibong Tebal 14300, Penang, Malaysia; dhamyaa.husam@muc.edu.iq (D.H.A.-N.); intan.sorfina@usm.my (I.S.Z.A.)

[2] Communication Engineering Department, Al-Mansour University College, Baghdad 10068, Iraq; laith.salman@muc.edu.iq

[3] Electronic Engineering Branch, Department of Electrical Engineering, University of Technology Iraq, Baghdad 30095, Iraq; 30095@uotechnology.edu.iq

* Correspondence: ashidi@usm.my

**Abstract:** The demand for bandwidth-critical applications has stimulated the research community not only to develop new ways of communication, but also to use the existing spectrum efficiently. Networks have become dynamic and heterogeneous. Receivers have received various signals that can be modulated differently. Automatic modulation classification (AMC) is a key procedure for present and next-generation communication networks, and facilitates the demodulation process at the receiver side. Under the presence of noise from the channel, the transmitter and receiver with its unknown parameters, such as carrier frequency, phase offset, signal power, and timing information, have become cumbersome because detecting the modulation scheme of the received signal is a complicated procedure. Two main methods, namely maximum likelihood functions and the signal statistical feature-based (FB) approach, are used for the automatic classification of modulated signals. In this study, a comprehensive survey of various modulation techniques based on FB approach is conducted. In this research, a number of basic features that are usually used in determining and discriminating modulation types were investigated. The classifier that was used in the discrimination process is studied in detail and compared to other types of classifiers to help the reader determine the limitations associated with the FB approach. Both classifiers and basic features were compared, and their advantages and disadvantages were investigated based on previous researches to determine the best type of classifier and the set of features in relation to each discrimination environment. This work serves as a guide for researchers of AMC to determine the suitable features and algorithms.

**Keywords:** automatic modulation classification; feature-based; likelihood-based; higher-order statistical; fast Fourier transform; continuous wavelet transform; decision tree; support vector machine; artificial neural networks; k-nearest neighbor

---

## 1. Introduction

With the recent advancements in telecommunication technologies, various bandwidth-critical applications have attracted the attention of users. This condition has required broadband data service providers to upgrade their network to fulfil the requirements of these applications. Thus, heterogeneous networks have evolved to support various types of data traffic on the basis of end user demands for supporting the existing voice and data services, and to meet the next-generation network requirements.

Managing the radio and other resources in a complex heterogeneous network requires real-time information extraction and processing.

With growing heterogeneity and dynamics, modulation recognition (MR) has attracted the attention of researchers in the last two decades. Radio frequency and microwave signals travelling in the air with different modulation types and frequencies fall in a huge band, and these signals should be identified and monitored for various applications. MR has been widely applied in various areas, such as mobile telephony, software and cognitive defined radio, military intelligence, communication jammers, surveillance, spectrum management, and communication reconnaissance [1].

At present, digital MR methods have witnessed a paradigm shift from manual operational systems to automatic ones because of several advantages. Manual modulation recognition (MMR) requires the measurement of parameters of intercepted signals to recognize modulation types. In MMR, four types of information are available for the search operator, namely, intermediate frequency (IF) time waveform, average and instantaneous spectrum of signal, sound, and signal and instantaneous amplitude. Manual analysis becomes problematic and inaccurate when the number of intercepted modulation types increases. In addition, this method requires experienced analyzers and does not guarantee reliable classification results. However, these shortcomings can be addressed via automatic modulation recognition (AMR). AMR is more powerful than MMR because it integrates an automatic modulation recognizer into an electronic receiver [2].

AMR refers to a process that helps in differentiating the modulation model from the received signal with unknown format and is conducted at the receiver [3]. AMR plays a significant role in cooperative and noncooperative communication. Cooperative communication is a means of transmission, wherein wireless users not only transmit their own information, but also repeat other users' information during their transmission to a common destination. Noncooperative communication is a type of classical multiple-access channel, wherein users directly send information to a common destination without repetition [4]. AMR is a challenging task, especially under noncooperative settings, because signal identification is difficult when the traits are unknown [5]. Similarly, blind AMR algorithms are considered to be complex because signal recognition is difficult due to the absence of prior knowledge of modulated signals. Thus, AMR algorithms have received considerable attention as a new and interesting research area [6].

Figure 1 illustrates the essential role of AMR in the receiver part of communication systems. The AMR block is composed of two parts: signal preprocessing and classifier modules. The preprocessing module estimates the synchronization parameters, such as the frequency offset of the received signal, timing recovery, and power. In the second part, signal disturbances (e.g., interference identification) are eliminated, resulting in enhanced performance [6]. In the literature, AMR techniques are classified into two classes, namely, likelihood-based (LB) and signal statistical feature-based (FB) approaches.

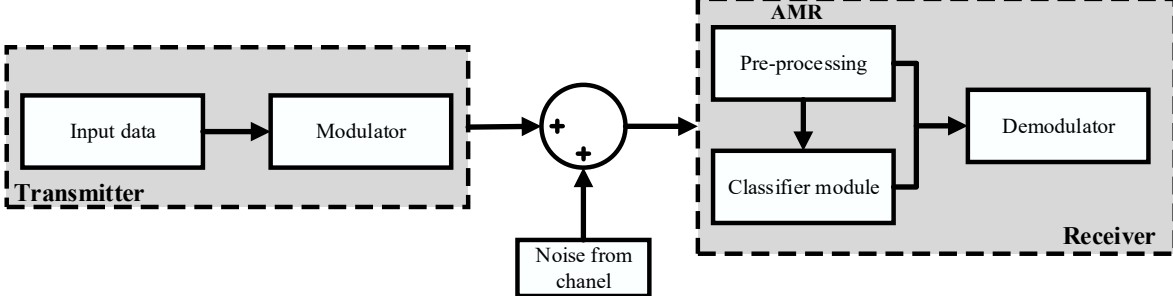

**Figure 1.** Communication system model that utilizes automatic modulation recognition (AMR).

In the LB approach, AMR is a multiple-composite hypothesis-testing problem. The LB approach is based on building a probabilistic model for the received signal, and decision is made to classify the modulation type by comparing the likelihood functions (LFs) or the likelihood ratio against a

threshold [7]. The average likelihood ratio test (ALRT) treats unknown quantities as random variables and computes the LF by averaging them. LB classification presents the classification of a combination of hypothesis and testing problems. Hypothesis estimates that each incoming signal has a modulation type. Generalized likelihood ratio test (GLRT) computes the probability density function (PDF) of incoming signals by applying maximum likelihood estimations (MLEs) based on unknown quantities. Then, the LF defines the most possible modulation type of the signal [8]. In addition to the two techniques, hybrid likelihood ratio test (HLRT), quasi ALRT, and quasi HLRT have been proposed in the literature. Several researchers have utilized the LB approach for MR [9,10]. The likelihood method shows optimal performance by increasing the accuracy of classification probabilities, but it has high computational complexity due to other computation likelihoods; nevertheless, it formulates the correct hypotheses, carefully selects the appropriate threshold values, and exhibits high sensitivity to modelling mismatches, such as timing, phase and frequency offset, and noise divergence [11]. When a closed-form solution exists, the computational complexity can make a classifier impractical. For the low signal-to-noise ratio (SNR) approximation of the LF, the quasi-ALRT algorithm provides near-optimal performance for recognizing FSK and PSK-modulated signals, but it fails in the case of quadrature amplitude modulation (QAM)-modulated signals. Although GLRT has benefits, it fails in finding nested constellations. HLRT with many unknown parameters has high time complexities. Complexity is decreased in quasi-HLRT classifiers, which rely on low complexity but accurate estimators.

The FB approach can be considered a mapping relationship that conducts mapping of time-series signals to the feature field, and these featured parameters are used for signal recognition [8]. Feature extraction is based on various aspects of signals, such as instantaneous amplitude, phase, frequency, wavelet transform (WT), zero-crossing intervals, statistical features (e.g., higher-order moments and cumulants), cyclic cumulants [8,12], and signal spectra [6]. In feature selection, the most relevant features must be utilized to improve the classification accuracy. After the signal features are extracted, they pass through classificatory decision to identify the modulation type [13]. The FB approach not only produces suboptimal performance but also has low computational costs. Therefore, this approach achieves near-optimal performance when it is systematically designed [12].

In addition to the LB and FB approaches which are usually used in AMR, there is another approach which is used in MR known as the 'distribution test' approach. In this approach, the empirical distribution of the modulated signal is used. The signal distribution is reconstructed using the empirical distribution, then the observed signals are analyzed using their signal distributions. If the theoretical distribution for the different modulation candidates is available, one will exist which is the best match for the underlying distribution of the signal we want to classify. The amount of equality between difference distributions is usually known as Goodness of Fit (GoF), which is an indication of how well the sampled data fit the reference distribution. Finally, the classification is completed by finding the assumed or hypothesized signal distribution that has the highest goodness of fit. A goodness of fit approach using Kolmogorov–Smirnov (KS) was suggested to solve the problem of modulation classification in different channels, like the AWGN channel, the flat-fading channel, and channels that have unknown phase and/or unknown frequency offsets [14].

In [15], the authors proposed a KS test for the classification of signals of higher order by comparing the difference between the empirical cumulative distribution function (ECDFs) from received signals and CDFs of the signal under different candidate modulation format to classify modulation type. The decision statistic results in the smallest value of the greatest distance between the two CDFs mentioned above [16]. Comparing the KS test with LB and FB methods, the KS test has less computational complexity and performs better at various channels than the LB method. It is more accurate than the FB method. However, the KS test is used only for the small datasets and is more complex than the FB method.

In this study, an in-depth survey is conducted on FB approaches for digital MR, which can be useful for readers who are looking for lucid information about the features and classifiers used in FB-AMR. The remainder of this paper is organized as follows: Section 2 presents the AMR signal model.

Section 3 discusses the FB approach and its features, which are used during classification. Section 4 investigates the classifier algorithms of pattern recognition (PR). Section 5 presents a performance analysis. Section 6 provides the conclusions.

## 2. AMR Signals

AMR is considered to be an intermediate step between signal detection and demodulation and has huge importance and applications in civil and military communication systems. In a heterogeneous network, a signal may be modulated by various modulation schemes, and a receiver might not be aware of the modulation scheme. Thus, AMR at the receiver not only recognizes the modulation scheme, but also successfully demodulates the signal. To accomplish these tasks, few features from the received signal are extracted and further processed to identify the type of modulation scheme. However, systems usually receive a modulated signal that is affected by noise and attenuated because of channel losses. Interference may also appear due to cross channel effect, making the task challenging and cumbersome.

### 2.1. Digital Modulation

In digital modulation techniques, an analogue carrier signal is usually modulated by a binary message code, and this process can be achieved by varying the physical characteristics of the carrier, such as amplitude, frequency, or phase, or a combination of them. Therefore, when the carrier amplitude varies, the modulation is named amplitude shift keying (ASK). When the carrier frequency changes on the basis of the message signal, the modulation is called frequency shift keying (FSK). When the carrier phase varies in accordance with the signal, the modulation is called phase shift keying (PSK). In addition, QAM is a combination of PSK and ASK [17]. Lower- and higher-order modulation schemes can be used for each category, such as binary frequency shift keying (BFSK), 16-QAM, and quadrature phase shift keying (QPSK) [18–26].

### 2.2. Communication Channel Model

Information signals in communication systems suffer from quality degradation when they are propagated from the transmitter to the receiver via a communication channel. The transmitted signals are attenuated and distorted in the channel. The signal loses some of its energy due to channel impedance, causing it to be attenuated [23]. Distortion changes the shape or form of the signal. This condition occurs when more than one signal arrives with different frequencies at the receiver side and when channel impairments exist. Another source of transmission impairment is random noise, which originates from several natural and artificial sources. This type of noise is classified as random because it is unpredictable and a crucial issue in the study of transmission impairments in communication systems.

The existence of these impairments in signal transmission makes it difficult to identify the modulation type used to transmit the information signal to the receiver. Additive white Gaussian noise (AWGN) is a common channel noise that usually affects the signal amplitude. Therefore, with the presence of noise, the received signal is expressed as [27]

$$r(t) = s(t) + n(t) \tag{1}$$

where $r(t)$ is the received signal waveform. The received signal consists of two uncorrelated components, namely, modulated signal $s(t)$ and noise $n(t)$ (i.e., introduced by the channel during transmission).

AMR algorithms introduced in the literature have utilized the information contained in the received signal, extracted from the baseband wave or intermediate frequency [28]. Generally, the signal received by the AMR for further processing can be expressed as

$$r(t) = \tilde{s}(t)e^{-j(2\pi f_c t + \phi_c)} + n(t) \tag{2}$$

where $f_c$ is the carrier frequency, $\varphi_c$ is the carrier phase, $\tilde{s}(t)$ is the baseband complex envelope of real signal $s(t)$ and is calculated as

$$\tilde{s}(t) = a(t)e^{-j(2\pi f(t)+\phi(t))} + n(t) \tag{3}$$

where $a(t)$ is the instantaneous amplitude of the signal and $f(t)$ and $\varphi(t)$ are the frequency and phase of the signal, respectively [29,30]. The modulated signal is represented through the real part of $\tilde{s}(t)$, as follows:

$$m(t) = \text{Re}\{\tilde{s}(t)e^{-j2\pi f(t)}\} \tag{4}$$

## 3. Overview of FB Approach for MR

The FB approach involves two main phases: feature extraction and classifier. The review of feature extraction for MR is presented in this section, and the review of the second phase on FB approach is discussed in Section 4.

Various digital signals enforce innumerable properties and characteristics, and appropriate and optimal feature(s) should be defined to recognize the signals. Therefore, such features that are sensitive to digital modulation schemes and insensitive to noisy channels and SNR variations should be selected [31]. No hard and fast rule exists for feature selection. Researchers have used various features and reported their outcome for MR. However, the literature primarily reports the usage of four common features namely, spectral, statistical, spectrum and constellation shape features. The review on each feature is presented in Sections 3.1–3.4, respectively.

### 3.1. Spectral Features for MR

Spectral-based features are frequency-based features that help in correlating physical attributes to perceptual attributes. Each feature is selected because of two reasons, namely, the mathematical calculation, which proves the goodness of the feature criteria, and the reported usefulness of the feature in previous research [29]. Spectral-based features of a signal was presented for the first time by Azzouz and Nandi at the end of the 20th century [32,33]. The parameters of instantaneous amplitude, phase, and frequency (which are presented as $a_n$, $\phi_{NL}$ and $f_N$, respectively) were used for signal classification. This process is an intuitive way of identifying the modulation class of incoming signals. FSK signals are defined by the fixed instantaneous amplitude. ASK signals are classified on the basis of their amplitude variations, and the PSK information is in their phase. They can be obtained in many ways, and a common method used is Hilbert transform [34]. Instantaneous amplitude A(t), phase ø(t), and frequency $f_N$ can be obtained using Equations (5)–(7), respectively, [8,35]:

$$A(t) = \sqrt{r^2(t) + \hat{r}^2(t)} \tag{5}$$

$$\varphi(t) = \tan^{-1} \frac{\hat{r}(t)}{r(t)} \tag{6}$$

$$f_N = \frac{1}{2\pi} \frac{d_{\varphi uw}(t)}{dt} \tag{7}$$

where r(t) is the real valued modulated signal and ř(t) is its Hilbert transform. Several spectral features are used for digital modulation identification, which are proposed in the literature and are tabulated in Table 1 [33].

**Table 1.** Spectral features used for modulation classification [33].

| Features | Mathematical Equation |
|---|---|
| **Maximum value of PSD $\gamma_{max}$ of the normalized centered instantaneous amplitude** | $\gamma_{\max} = \frac{\max\|DFT(A_{cn}(i))\|^2}{N_s}$, where DFT is the discrete Fourier transform of the modulated signal, $N_s$ is the sample number, $A_{cn} = \frac{A_i}{\mu_A} - 1$, $A_i$ is the $i^{th}$ instantaneous amplitude and $\mu_A$ is the sample mean |
| **Standard deviation of the absolute values of the centered nonlinear components of instantaneous phase $\sigma_{ap}$** | $\sigma_{ap} = \sqrt{\frac{1}{N_c}\left(\sum\limits_{A_n(i)>A_t} \varphi_{NL}^2(i)\right) - \frac{1}{N_c}\left(\sum\limits_{A_n(i)>A_t} \|\varphi_{NL}(i)\|\right)^2}$, where $N_c$ is the number of sample(s) in $\{\phi_{NL}\}$ for $A_n(i) > A_t$, where $A_t$ is the threshold value of $A_n(i)$ when the filter provides the minimum amplitude of the signal sample due to high noise sensitivity and $\phi_{NL}(i)$ is the nonlinear component of the $i^{th}$ instantaneous phase of the sample. |
| **Standard deviation of the absolute value of the normalized centered instantaneous amplitude in the nonweak segment of signal $\sigma_a$** | $\sigma_a = \sqrt{\frac{1}{L}\left(\sum\limits_{A_n(i)>t_{th}} a_{cn}^2(i)\right) - \frac{1}{L}\left(\sum\limits_{A_n(i)>t_{th}} \varphi_{cn}(i)\right)^2}$, where $L$ is the length of the non-weak value and $t_{th}$ is the threshold value of the non-weak signal. |
| **Standard deviation of the direct value of the centered nonlinear component of the direct instantaneous phase in nonweak segment $\sigma_{dp}$** | $\sigma_{dp} = \sqrt{\frac{1}{N_c}\left(\sum\limits_{A_n(i)>A_t} \varphi_{NL}^2(i)\right) - \frac{1}{N_c}\left(\sum\limits_{A_n(i)>A_t} \varphi_{NL}(i)\right)^2}$, where all parameters are similar to $\sigma_{ap}$ but differs in the absence of the absolute operator in the nonlinear component of the instantaneous phase. |
| **Standard deviation of the absolute value of the normalized centered instantaneous amplitude of signal segment $\sigma_{aa}$** | $\sigma_{aa} = \sqrt{\frac{1}{N_c}\left(\sum\limits_{i=1}^{N} A_{cn}^2(i)\right) - \frac{1}{N_c}\left(\sum\limits_{i=1}^{N} \|A_{cn}^2(i)\|\right)^2}$, where $A_{cn}$ is the normalized and centered instantaneous amplitude of the incoming signal at the time instant. |
| **Standard deviation of the absolute value of the normalized centered instantaneous frequency of signal segment $\sigma_{af}$** | $\sigma_{af} = \sqrt{\frac{1}{N_c}\left(\sum\limits_{A_n(t)>A_t} f_N^2(i)\right) - \frac{1}{N_c}\left(\sum\limits_{A_n(t)>A_t} \|f_N(i)\|\right)^2}$, where $f_N$ is the normalized frequency. |
| **Kurtosis of the normalized centered instantaneous amplitude $\mu_{42}^a$** | $\mu_{42}^a = \frac{E\{A_{cn}^4[n]\}}{\{E\{A_{cn}^2[n]\}\}^2}$. |
| **Kurtosis of the normalized centered instantaneous frequency $\mu_{42}^f$** | $\mu_{42}^f = \frac{E\{f_N^4[n]\}}{\{E\{f_N^2[n]\}\}^2}$. |

Previous studies have presented several FB approaches on time and frequency domains. Table 1 presents the specific purpose for each feature. The maximum power of spectral density (PSD) depends on the information conveyed through a signal envelope, such as M-ASK and M-QAM for digital modulation, and the $\gamma_{\max}$ value must be nonzero. When the modulated signal has a constant amplitude, such as M-FSK and M-PSK, the $\gamma_{\max}$ value must be zero. Therefore, $\gamma_{\max}$ can be used to discriminate between M-ASK and M-QAM in addition to constant amplitude M-FSK and M-PSK digital modulation schemes [4,31,33,35–39]. The average value of $\gamma_{\max}$ is used to discriminate among continuous phase FSK, Gaussian FSK, and Gaussian minimum shift keying (GMSK) [40]. In addition, several studies have used these features to discriminate the order of modulation schemes, such as M-ASK and M-FSK [41].

Features $\sigma_{aa}$ and $\sigma_a$ are used to detect the variant of modulated signal amplitude. These features are similar to $\gamma_{\max}$. Therefore, these features are used to classify the order of M-ASK and M-QAM [42]. In [35,38,43], $\sigma_{aa}$ is used to discriminate between the orders of ASK. $\sigma_a$ feature is helpful for recognizing ASK from PSK [31]. Features $\mu_{42}^a$ and $\mu_{42}^f$ represent the distribution of amplitude and frequency, respectively. $\mu_{42}^a$ is used to separate FSK/PSK from ASK/QAM, whereas $\mu_{42}^f$ is used to separate FSK from PSK [8,36].

Feature $\sigma_{af}$ highlights the variation in signal frequency; therefore, it can be used to recognize the order in FSK [7,35,42]. Features $\sigma_{ap}$ and $\sigma_{dp}$ characterize the variations in a signal's instantaneous

phase. $\sigma_{ap}$ is used to discriminate the order of PSK [4,44], and it distinguishes among ASK2, ASK4, PSK2, and PSK4, Therefore, the gap mainly depends on the absolute phase information to discriminate the modulation formats. $\sigma_{dp}$ uses the direct phase information to distinguish the modulation types (containing the direct information phase) from those that do not have the direct information phase, such as distinguishing among ASK2, ASK4, and PSK [35,43].

Although these features are simple to extract, they are sensitive to noise and can produce estimation errors. Furthermore, the extraction of instantaneous information fully depends on thresholds that should be set normally in advance.

## 3.2. Statistical Features for MR

Some types of modulation schemes contain information in the amplitude and phase spectra, which are known as complex signals, and are represented as in-phase and quadrature components. In this context, statistical features are useful for eliminating the background noises caused by channel environments and the presented fluctuation [7,29]. Statistical features include higher-order statistical (HOS) features, such as moment and cumulants, which are widely used for feature discrimination on ASK, PSK, and QAM modulation types. The main strengths of higher-order statistics are robustness against phase rotation, the capability to remove noises, and reflection of higher-order properties of the signal [45]:

Moment is a general concept of the probability distribution moment functions. Moment is a method for quantitatively measuring the shape of a function [31]. In 1992, Soliman and Hsue [46] introduced a third-order moment-based modulation classifier for classifying the order of M-PSK; they found that the moment of the phase signal in shared white Gaussian channel is gradually monotonically increasing to M-PSK order. Thus, this feature can be used to classify modulation types. The $K^{th}$ order moment $K(r)$ is determined theoretically as [33,46]

$$k(r) = \frac{1}{N} \sum_{n=1}^{N} \varphi^k(n) \tag{8}$$

where $\phi^k(n)$ is the phase for the $n^{th}$ sample signal received and $N$ represents the total number of samples. The calculation of different $k^{th}$ moments of received signal $r = r[1], r[2], \ldots, r[N]$ [33,46] can be expressed as

$$Moment_{xy}(r) = \frac{1}{N} \sum_{n=1}^{N} r^x[n] \times r^{*y}[n] \tag{9}$$

where $x + y = k$ and $r^*[n]$ is the complex conjugate of $r[n]$ [33].

Higher-order cumulants (HOCs) are the second statistical features investigated after the moments. Fourth-order cumulants of the complex signal value are suggested by [47] as the feature for classifying modulation types M-PAM, MPSK, and M-QAM when the carrier phase and frequency offsets are unknown. The HOC equation is obtained from a combination of two or more of higher-order statistical moments (HOMs) [45] as in $C_{42}$, which is expressed as

$$C_{42} = cum[r(n)r(n)r(n)r^*(n)] = M_{41} - 3M_{20}M_{21} \tag{10}$$

where c42 is the fourth-order moments and cumulants, r[n] is the complex-valued stationary random process, r*[n] is the conjugate of r[n], M20M21 are the second-order moments, and M40 is the second-order moment. HOC features are robust to noise, especially when cumulants higher than the second order are used. HOCs of demodulated signals are approximately equal to those of transmitted signals [48,49].

The cumulant values can be used for modulation discrimination in two ways. The first method is by creating hierarchical schemes that precisely discriminate the order and type of the digital

modulation (M-ASK, M-FSK, M-PSK, M-QAM). The second utilization of cumulants is introduced in [50] by comparing the estimated value to the real value. HOS features are used to classify M-PSK and M-QAM modulation types, which are robust against AWGN [12,40,48,51–55]. Furthermore, the multipath channel effects can be easily modelled using the HOS features [12,31,56,57], which are robust to frequency, phase offset, and timing errors [52,53,58]. In [48,49], the extracted features from ratio and absolute of HOC are used as a characteristic parameter for classifying between M-FSK and M-PSK, M-ASK, and 16 QAM, and between M-PSK and M-ASK [59]. In [53], the authors used fourth-order cumulants to recognize M-PSK. They divided their algorithm into two steps. In the first step, the fourth-order zero conjugate cumulants of backward difference samples of the receiver signals were used to identify QPSK and OQPSK. In the second step, the fourth-order zero-conjugate cumulants of the received noisy signal and their squares were used to identify QPSK, 8 PSK, and 16 PSK in the presence of frequency and phase offset. In [60], the authors used HOMs as features to recognise nine classes of modulation types. Second-order moments were utilized for the intraclassification of PAM.

### 3.3. Transform Features for MR

The spectral characteristics of an FSK signal is proven to be recognized from PSK and QAM signals and to split various M-ary FSK schemes [61]. Spectrum analysis-based classification is a scheme that is used to identify MFSK modulation schemes by using fast Fourier transformer classifier (FFTC) and DFT via fast Fourier transform (FFT) [62]. In [61,63], the features were extracted from the FFT and then compared with the threshold values, which were used to discriminate the types of M-FSK. This technique can be used in real time and is useful in reducing the input noise and decimating the input signal to the automatic modulation classification (AMC) [64].

WT provides an environment for analyzing signals under different frequencies and different resolutions. Hence, particular features of signals are only revealed in this domain or cannot be extracted using other forms of transform [30]. WT can be categorized into continuous wavelet transform (CWT) and discrete wavelet transform (DWT). CWT is used to overcome the resolution problem, and short-time Fourier transform can be alternatively used [65]. CWT of received $S$ is defined as the integral of $S(t)$ times the conjugate transpose of wavelet function $\Psi_{a,x}(t)$ over a time duration, as follows:

$$CWT(a, \tau) = \frac{1}{\sqrt{|a|}} \int_{-\infty}^{\infty} S(t) \psi_{a,\tau}^* \left( \frac{t - \tau}{a} \right) dt \tag{11}$$

where $\Psi^*$ is its complex conjugate, $a \neq 0$ is the scale, and $\tau$ is the translation variable.

The wavelet function has been extensively discussed in the literature, such as by Morlet, Haar, and Shannon, and the most applied wavelet within the modulation field is Haar wavelet because of its simplicity and computation convenience. Haar used a simple rectangular waveform as the reference wavelet for a classifier to decompose the received modulating signal [66].

Most studies on the field of wavelet domain-based features have focused on the histogram computation of CWT and/or DWT wavelet coefficient of demodulated signals. The number of peaks in histograms is used for identifying different types of digital modulation [67–70]. The histogram peaks in WT magnitude, mean, and variance of normalized histogram are utilized for identifying the type of digital modulation [71]. In [72], the author analyzed the CWT instantaneous features (mean, variance, and central moment values) for recognizing M-ASK, M-PSK, M-FSK, M-QAM, OOK, and MSK. In [73], the histogram peaks in WT magnitude, mean, and HOM of normalized histogram were adopted as features for digital modulation classification. The wavelet variation coefficient difference and similarity measurement functions from statistics were used as the features to classify MASK, MFSK, MPSK, and MQAM [74].

In practical scenarios, modulating signal power exhibits uncertainty. Therefore, an optimal solution that covers the uncertainty should be determined. In 1994, Gardner firstly introduced cyclostationary analysis for the received signal to investigate the diversity in the spectrum appearance of all modulation

types [75]. Random signals are classified as cyclostationary when the HOMs of their envelope are periodic [33]. Each modulation type has different periodic spectrum patterns [45]. A common signal *x(t)* of a sinusoidal function is considered to be a cyclostationary feature or second-order periodicity when its cyclic autocorrelation function can be expressed as

$$R_x^\alpha(\tau) = \lim_{T \to 0} \frac{1}{T} \int_{-T/2}^{T/2} X(t + \frac{\tau}{2}) \times X^*\left(t - \frac{\tau}{2}\right) e^{-j2\pi t} \tag{12}$$

where $R_x^\alpha(\tau)$ is a Fourier series coefficient called cyclic autocorrelation function, *X(t)* is the signal itself, $X^*(t)$ is its conjugate and *T* is the time interval. The spectral correlation function can be defined as the cyclic spectrum. Cyclic autocorrelation can be obtained by using Fourier transform, as follows:

$$S_x^\alpha(f) = \int_{-\infty}^{\infty} R_x^\alpha(\tau) e^{-j2\pi t} dt \tag{13}$$

where $\alpha$ and *f* are the cyclic and spectrum frequencies, respectively [76].

In 2003, Dobre used higher-order cyclic cumulants to identify ASK, PSK, and QAM [77]. In [78,79], cyclostationarity-based classification of M-FSK was performed using the first-order cycle moment corresponding to zero at frequencies other than cycle frequencies (CFs). Here, the number of first-order CFs was the feature extracted from the incoming signal and utilized to determine the modulation order, *M*, of an FSK received signal. The number of first-order CFs detected in the received FSK signal was used as the discriminating feature for modulation order recognition. Therefore, the first CF was equal to 2FSK, 4FSK, and 8FSK. This classifier is independent of the information estimation of timing, frequency, and channel information. In [80], a second-order cyclic moment-based symbol period estimation algorithm was introduced to identify M-ary FSK without the requirement of carrier and channel recovery. A fourth-order cyclic was used to identify ASK, 2PSK, and QPSK, and the results showed the robustness of the algorithm against noise.

Cyclostationarity does not require previous knowledge of carrier phase, carrier frequency, and time offsets; therefore, it is widely adopted to blind AMC schemes [45,81,82]. Such a feature exhibits a periodic robust cyclic frequency, which enhances the classification outcomes [45].

### 3.4. Constellation Shape Features for MR

Constellation shape, which refers to the total number of points and its locations, is used to investigate the geometry shape of the constellation diagram of modulating signals, such as PSK and QAM signals. Each location has distinct distance and phase with respect to the origin point and is used by Mobasseri [83] to transfer the phase–amplitude distribution to 1D distribution. In [84,85], the authors adopted constellation features to identify M-QAM. However, the shortcomings of this method were its sensitivity to noise and high SNR requirement in identifying higher-order modulation. In [86], the authors improved the quality of M-QAM identification by combining the constellation WT characteristics of QAM signals. This feature was robust to AWGN. However, the SNR should be greater than four to achieve good performance.

### 3.5. Critical Analysis of the FB Approach for MR

Table 2 presents the review summary based on the discussions in Sections 3.1–3.4. The selected features, which are used in the literature to discriminate the modulation types, should be further sensitive in terms of discrimination and insensitive to noisy channels and SNR variations with reduced computational complexity. Spectral features have low complexity in terms of implementation. However, they are sensitive to additive noises. Higher-order statistics have high resistance to AWGN and multipath channels, and are further sensitive in discriminating between modulation schemes, such as M-PSK and M-QAM. Cyclostationary features have good resistance to noise at low SNR, but have high complexity and cannot discriminate a large set of modulation schemes. FFT features are

robust at low SNR and further sensitive in discriminating M-FSK modulation schemes. The wavelet feature is suitable for discriminating modulation schemes at high SNR and requires many samples. The constellation feature is further sensitive to noise and requires high SNR to achieve good performance.

**Table 2.** Summary of the advantages and disadvantages of key features used.

| Reference(s) | Key Features | Modulation Set | Advantages | Disadvantages |
|---|---|---|---|---|
| [35] | Spectral features | 2ASK, 4ASK, 2FSK, 4FSK, 2PSK, 4PSK, 8ASK, 8FSK, and 8PSK. | • Good discrimination of modulation types | • Require high SNR<br>• Require many samples |
| [38] | Spectral features | 2ASK, 2PSK, 2FSK, 4ASK, 4PSK,16QAM, and 4FSK | • Good discrimination of modulation types for digital modulation schemes at low SNR | • Small set digital modulation schemes<br>• Only discriminate low-order modulation schemes |
| [43] | Spectral features | 2ASK, 4ASK, 2FSK, BPSK, and QPSK | • Good recognition of digital modulation at low SNR | • Small set digital modulation schemes |
| [12] | Sixth and fourth-order cumulants | BPSK, QPSK, QPSK, 16-QAM, 64-QAM | • Good recognition of modulation schemes by using AWGN and multipath fading channel in real time | • High recognition at high SNR and increased number of samples |
| [31] | Eighth-order moment and eight-order cumulant | 16QAM, 64QAM, and 256QAM | • Robust to AWGN | • Poor recognition of high-order modulation schemes (64QAM and 256 QAM) |
| [48] | Second and fourth cumulants | 2ASK, 4ASK, 4PSK, 2FSK, 4FSK, and 16QAM | • Reduced computational complexity<br>• High discrimination of modulation schemes | • Poor recognition of M-FSK<br>• Require high SNR |
| [39] | 2nd, 4th, and 8th-order cumulants | BPSK, QPSK, QAM, 16QAM, and 64QAM | • Good discrimination between M-PSK and M-QAM | • Recognition rate is enhanced with the increase of SNR and number of samples |
| [51] | HOCs | BPSK, QPSK, 8-PSK, 16-QAM, 64-QAM, and 256-QAM | • High discrimination between (M-PSK and M-QAM) when the samples are reduced | • Require high SNR and cannot recognise between 64QAM and 256 QAM at low SNR |
| [53] | HOCs | QPSK, OQPSK, 8-PSK, and 16-PSK | • Good discrimination between M-PSK modulation schemes | • Recognition rate is enhanced with the increase of SNR and number of samples |
| [54] | HOCs | BPSK, QPSK, QAM, 16QAM, and 64QAM | • High discrimination between M-PSK and M-QAM at low SNR | • Require many samples |
| [76] | Cyclic frequency domain | 2FSK, 4FSK, 8FSK, BPSK, QPSK, MSK, and 2ASK. | • Robust at low SNR | • Computational complexity |

**Table 2.** *Cont.*

| Reference(s) | Key Features | Modulation Set | Advantages | Disadvantages |
|---|---|---|---|---|
| [78,79] | Cyclic frequency domain | 2FSK, 4FSK, 8FSK | • Good discrimination between the M-FSK | • Poor discrimination of M-FSK at low SNR<br>• Small set of modulation schemes |
| [81] | Cyclostation-ary | M-QAM | • Robust to timing, phase, frequency offsets, and phase noise | • Discriminate only three types of M-QAM<br>• High identification of modulation schemes at high SNR |
| [62] | FFT | 2FSK, 4FSK, 8FSK, 16FSK, 32FSK | • Good discrimination of M-FSK at low SNR | • Only MFSK signals are considered.<br>• Cannot discriminate additional modulation types from the frequency spectrum |
| [63] | FFT | 2FSK and 4FSK | • High discrimination between M-FSK at lower SNR | • Requires many samples |
| [65] | WT | BASK, BFSK, and BPSK | • Robust at low SNR | • Discriminate a small set of modulation schemes |
| [66] | WT | PSK, QAM, FSK, and ASK | • Good discrimination of modulation schemes | • Requires high SNR |
| [70] | WT | 4QAM, 16QAM, and 64QAM | • Good discrimination of M-QAM | • Recognition rate is increased with increasing SNR and number of symbols<br>• Recognition rate is low at low SNR. |
| [84] | Constellation | 4QAM, 16QAM, 32QAM, 64QAM, 128QAM, 256QAM | • Shows good discrimination of M-QAM | • Cannot effectively recognise the higher-order QAM signal at lower SNR |
| [85] | Constellation | 16QAM, 32QAM, and 64QAM | • Shows good discrimination of M-QAM | • Constellation identification normally requires high SNR |

## 4. Overview of the Type of Classifiers Used for MR

The step after feature extraction in AMC is the identification of the type of received signal [7]. Several common approaches, such as decision tree (DT), artificial neural networks (ANNs), support vector machine (SVM), k-nearest neighbor, and combinations of artificial intelligence techniques, have been used for classification. ANN and SVM are supervised machine learning, whereas clustering classifier belongs to unsupervised machine learning [83–85]. Researchers have used optimization techniques to select the dominant or significant features from the extracted features for improving the recognition accuracy of classifiers. They have considered AWGN to be the noise for AMC (because of its simplicity). However, few researchers have considered fading channels in AMC. The details regarding each classifier are provided in the subsequent section.

## 4.1. DT

A DT is a decision support tool that uses a treelike model of decisions and their possible consequences (i.e., modulation scheme type) [7]. DT, as a classifier for AMC, follows sequential decision-making procedures in which specific thresholds are selected to separate the modulation type and orders. The decision making in any stage depends on the previous stage decision (test), which is taken on the basis of predefined threshold values that mainly influence the overall performance [31]. A DT algorithm can handle a multiclass problem. Each stage in the DT evaluates a specific feature. Any additional modulation involves a new decision by making new branches to the tree. Therefore, DT is considered to be less complex with acceptable performance than PR techniques [45]. However, DT consumes considerable time during design and optimization [87].

In [35], instantaneous FB DT was used to classify the digital modulation schemes. However, M-PSK and the required increased number of samples were not identified to enhance classification accuracy. Previous studies [48,49,52,53,58,60,88] used HOS as the features for DT, whereas [59] combined instantaneous and HOS as the features for DT. In [40], DT used a hierarchal hybrid by combining FB and LB, where the first stage was to discriminate among MFSK, PSK, and QAM subclasses depending on the variance threshold of instantaneous amplitude. The second stage in each DT depended on predefined threshold to recognize the types of FSK and fourth HOCs for differentiating M-PSK and M-QAM. For probability correct classification (PCC), MLE was used to identify the order of PSK, and QAM was utilized to enhance the classification accuracy. In [66,71,74], and [89], the authors used wavelet domain as the features for identifying the digital modulation.

## 4.2. ANN

ANN is a classifier approach commonly used in AMC. ANNs are motivated by the central nervous system and are widely used in PR and machine learning domain [90]. NNs include various layers, i.e., the input layer, the output layer, and different numbers of hidden layers in between. Each layer includes several nodes that are called neurons, which simulate human brain processing strategies. The nodes from previous layer are interlinked with corresponding nodes in the following layer on the basis of weights. Weights are responsible for mapping the input to the output [91]. Furthermore, ANN classifiers can be either supervised or unsupervised networks. The discrimination between the two networks is based on the prior existence of training data for learning purposes. In supervised ANNs, two data sets are used: training and test sets. In unsupervised networks, ANN clustering inputs the data into different clusters for training [92].

Several problems exist in ANNs that affect their performance. ANNs have limited generalization capability at low SNR. In [43], the classification accuracy for ANN was enhanced to identify the digital modulation signal at low SNR. However, this method only covered lower-order digital modulation. In [76], the cyclic frequency domain and backpropagation neural network (BPNN) were used as input attribute and classifier, respectively. This method improved the accuracy at low SNR but required considerable training. In [93], a discrete wavelet NN group was used to classify the digital modulation types for improving higher-order recognition. In this method, a single NN was distributed into multiple ANNs, which increased the complexity of the algorithm. Optimization techniques for enhancing the learning process and reducing the complexity are reported to be helpful. In [72,94], principal component analysis (PCA) was used to reduce the complexity of ANNs and improve the classification accuracy at low SNR. BPNN was used in conjunction with reduced feature vectors through PCA to reduce training time and computational complexity and obtain higher-order MR at low SNR. In [57], the artificial bee colony (ABC) technique was used to improve the performance of ANNs in multipath fading channels. ABC was used to find the optimal weight of ANNs. However, this method required high SNR to obtain good accuracy.

An interesting work on deep neural network (DNN)-based AMC was reported in [95], which used a DNN structure with three hidden layers. Twenty-one features from data samples were fed to the classifier. The proposed classifier produced the probabilities of each modulation class at each

node by utilizing a Soft Max layer at the output layer. The proposed model was evaluated for AMC under AWGN and Rican channel environments by considering Doppler's frequencies and SNR range. The results showed that 90% classification accuracy was achieved.

A recent development was the use of deep DNN to identify the signals in fiber-optic network. The algorithm used by [96] starts by applying the constant modulus algorithm (CMA) equalization then finding the signals amplitude histograms (AH) and feeding the data to the DNN. The non-data-aided (NDA) modulation format identification (MFI) was used for identifying three types of modulation techniques and achieved a 100% accuracy over a wide range of optical signal to noise ratio (OSNR).

The authors of [97] proposed a deep learning (DL)-based intelligent constellation analyzer that could realize modulation recognition as well as OSNR estimation. They utilized CNN as DL models and treat constellation map as original constellation distribution without knowing any other parameters of the M-PSK and M-QAM signals to be identified.

*4.3. SVM*

SVM is a type of supervised machine learning that can be used for classification. Although ANNs are widely used for AMC, they have certain limitations that SVMs can overcome, i.e., training may degrade the performance, resulting in overfitting and/or local minimum [33]. SVMs can solve overfittings at low SNR conditions [90]. In SVMs, a kernel function is utilized for the nonlinear mapping of features from the input to the feature domain. The common kernel functions used are radial basis function, multilayer perception, and polynomial functions [90]. SVMs are used for classification, especially on multiclass classification. The techniques used for such cases are either repetitive banded support vector machine (BSVM) or multiclass support vector machine (MSVM). In BSVM, SVMs are used to classify the first class from all other classes and then reused for classifying the second class from the rest until the last class is reached. In MSVM, a higher-dimension feature space is used [33,45,98]. In [99], genetic programming (GP) was adopted to select the best features from HOC features for M-QAM classification. However, this method was complex and covered small types of modulation schemes.

In [29], particle swarm optimization (PSO) was used to configure the kernel parameter for feature selection and digital modulation type classification for improving the performance of SVMs. HOS and wavelet were used to identify different digital modulation types. This method improved the recognition accuracy for different modulation schemes at low SNR although it was computationally expensive. In [28], the researchers used spectral features based on MSVM classifier to identify six digital modulation schemes, and this method was robust to signal length under various SNR ranges. In [30], the recognition accuracy with minimum SNR was improved by using a hierarchal classifier based on a binary tree SVM classifier for discriminating digital modulation. This method required few computations but only covered few modulation schemes. In [34], the recognition accuracy for digital modulation schemes was enhanced under various SNRs. Redundant features were removed, and the remaining features were trained using an SVM classifier. However, the determination of M-ASK at low SNR remained challenging in this work.

For the identification of optical signals, SVM was applied to estimate the OSNR and then classify the modulation format. However, these methods exploit statistical features of signals that are directly detected. Therefore, it cannot be applied to dispersion-unmanaged coherent optical-systems [100]. In [101], an SVM-based approach is applied to jointly estimate the modulation and the OSNR, which relies on the CDF of the received signal's amplitude in combination with SVM-based classification and regression for coherent optical receiver. The average modulation classification rate was 99%.

*4.4. KNN*

KNN is a simple classifier and is defined as a nonparametric approach that does not require information about data distributions. The fundamental principle of KNN is the distance between samples; the first sample is called the trained sample, whereas the second sample is the test sample.

A KNN classifier calculates the distances of the test sample with respect to the class in the training sample [90]. In [102], eight digitally modulated signals, namely, 2FSK, 4FSK, MSK, BPSK, QPSK, 8PSK, 16QAM, and 64QAM, were classified using KNN. KNN has various disadvantages, such as nonparametric, lazy learner, cannot determine parameter k, and computationally greedy algorithm [12]. GP was used in conjunction with KNN in [54,103]. In [103], BPSK, QPSK, QAM16 and QAM64, which are included in the IEEE 802.11a standard, were considered. The classification process was divided into two stages to improve the classification accuracy. Two trees were created in each stage for the classification of BPSK, QPSK, QAM16, and QAM64 modulation schemes. A single tree was created at the first stage to classify BPSK, QPSK, and QAM16/QAM64. Another tree was created at the second stage to classify QAM16 and QAM64. In [90], the performance of KNN was compared with that of another classifier that uses cyclostationarity features. The results showed that the performance and computational complexity of KNN were better than those of SVMs at low SNR.

### 4.5. Clustering Algorithms

Clustering algorithms for modulation classification have been used for analogue communication in the literature, which provides a comparison among clustering algorithms for analogue MR [104]. Clustering algorithms, which belong to a class of unsupervised machine learning, are used for data classification. Clustering algorithms aim to find the similarities in the dataset, and the clusters are formed on the basis of the similarities. For instance, assuming a set of animals, such as dog, cat, parrot, sheep, and pigeon, we can create two clusters from the animal data set on the basis of whether the animal can fly or not [84]. Although clustering algorithms are not popular for AMR, few clustering algorithms, such as k-means [85], fuzzy c-means [83], and subtractive clustering [84,105], have been used in the literature. An MQAM MR method was discussed in [106], thereby proposing a cyclic approximation method for estimating the carrier frequency and a method for subtractive clustering with a clustering radius. The simulation results demonstrated that such algorithms acquired 100% classification success rate for 4-QAM, 16QAM, 32QAM, and 64QAM at 10 dB SNR. In [107], two threshold sequential algorithmic schemes and PR were proposed to identify QAM and PSK. Classification was conducted by utilizing the constellation of the received signal through fuzzy clustering, and hierarchical clustering algorithms were used for classification.

### 4.6. Complexity Analysis for Classifiers

Some studies discussed the complexity analysis of the classifier used in MR in different approaches. For the ANN classifier in [38], the rough sets used for the selected features with the NN reduced the network complexity. A combination of the rough sets and neural network has smaller training time. This method can remove the redundant features from the training samples, and simplifies the structure of the ANN classifier. In [72,94], the features subset selection using principal component analysis (PCA) is used to reduce the complexity of the used neural network through the selection of the best features.

Similarly, in SVM classifiers, some researchers were also interested in reducing the complexity. In [29], the number of features extracted from the dataset includes 27 different features, so the complexity of the computation and runtime will be increased. Therefore, a feature subset selection process is needed to reduce the SVM complexity through the use of PSO. In [30], the hierarchical SVM classifier was used and has considerably low complexity compared to the structure of the OAO approach. This is because the number of the required SVMs in the proposed tree scheme for recognition of seven different modulation types is equal to six, while the OAO approach needs $7(7-1)/2 = 21$ SVMs. In [34], the author reduced the computation complexity of the SVM classifier through using the technique of the rough set. This technique works by selecting the best features from the original set. Four features were chosen from 25 features, which eliminates redundant information in the original features and reduces the complexity of the training process.

DT techniques are simple to design and implement, and do not need training of the classifier as in PR techniques. In [48,63], the researchers used two and three features, respectively, based on

the DT algorithm and hence reduced the computational complexity. In [40], a hybrid approach was used that implements both the FB and LB modulation classification for the efficient classification of multiple linear and nonlinear modulation formats. This hybrid classification approach is helpful in both reducing time and reducing computational complexity.

Reducing the complexity in KNN techniques was the aim of some researchers. In [54], the GP generated super features from the dataset based on a single-stage strategy that leads to the reduction of the computational time. In [90,108], the authors compared the KNN classifier to other classifiers, such as SVM and ANN, and found that the KNN has less computational complexity than SVM and ANN.

Clustering, which is another classifier, has drawn attention of researchers trying to reduce the computational complexity. In [84], the subtractive clustering algorithm, in which the number of cluster centers is adaptive, was used. This is greatly reduces the computational complexity. In [85], the K-means clustering method, which needs to calculate the number of clustering center points for each hypothesis, was used, leading to an increase of computational complexity.

*4.7. Critical Analysis on Different Classifiers for MR*

Table 3 presents the summary of the review presented in Sections 4.1–4.5.

**Table 3.** Summary of the merits and demerits of classifiers.

| Reference(s) | Classifier(s) | Merits | Demerits |
|---|---|---|---|
| [35,52,53,58,59,63, 71,74,88,89] | DT | Simple implementation. This method can accommodate many modulations by adding additional decision branches | Sensitive to noise when the threshold changes |
| [43] | ANN | High classification rate. Reduced computational complexity | Ineffective when a large number of input features are used. Small data set |
| [76] | ANN | Improved accuracy | Requires long training time for computation. Small-order modulation types |
| [72,94] | ANN | Reduced computational complexity through PCA. Improved accuracy | Small-order modulation types |
| [99] | GP with SVM | Improved classification performance | Two modulation types, namely, 16QAM and 64QAM, are used |
| [29] | PSO with SVM | Robust to noise. Good classification | Computationally complex |
| [34] | Rough set theory with SVM | Reduced computational complexity | Poor discrimination of M-ASK at lower SNR |
| [54] | GP with KNN | Reduced complexity | Small set of modulation schemes |
| [84,85] | Clustering | A priori information about signal is not required. | Requires high SNR |

As shown in Table 3, machine learning performs efficiently on small data sets and requires low implementation costs. However, the complexity of ML increases and leads to performance degradation when large data sets and complex features are used for recognition. ML cannot be applied in real-time environments. SVM and KNN classifiers can produce higher classification rates than ANNs. KNN classifiers are preferred because they are less complex than SVMs. The performance of DT is lower than that of ANN, SVM, and KNN. However, DT is simple to design and implement and can classify a wide range of modulation schemes by adding many decision points without retraining the classifier. Clustering classifiers are sensitive to noise but have high computational complexity.

## 5. Performance Analysis

This section presents a comparative analysis of the reviewed studies to resolve the main study question, i.e., what are the appropriate AMC techniques for different modulation types in terms of overall performance by using different feature extraction methods and types of classifiers? Comparison of different classifiers is a complicated work. Researchers have developed various algorithms for AMC under different signals and conditions. Consequently, the performance of different classifiers cannot be compared when the modulation scheme is different. Furthermore, the feature set used by each system must be mentioned to achieve unbiased and reliable comparison. The benchmark of performance comparison is SNR, and accuracy rate is used to assess the classification system performance. Table 4 presents a lucid and comprehensive comparison of various works on AMC with their key features as the input and SNR and accuracy as the output parameters.

As shown in Table 4, researchers have used various techniques in terms of feature selection, classifier types, and modulation set, and have evaluated their proposed methodologies under various SNRs. Three main conclusions can be derived as follows:

1.  Selection of appropriate features can improve system robustness towards noise effects and can be sensitive in terms of discriminating the modulation schemes, thereby enhancing the performance.
2.  Features and classifier types with reduced complexity are crucial for enhancing the performance of classifiers.
3.  Recognition is difficult when higher-order modulation types are used for higher-order QAM at low SNR.

**Table 4.** Summary of related works for feature-based (FB) modulation classification.

| Reference and Year | Classifier Types | Key Features | Modulation Set | SNR (dB) | PCC % | Advantages | Disadvantages |
|---|---|---|---|---|---|---|---|
| [99] 2011 | GP-SVM | HOC | 16QAM, 64QAM | 10 | 99.8 | • Performance enhancement | • Requires high SNR • Limited to modulation types used |
| [29] 2012 | SVM with PSO | Combines features: spectral, higher-order statistical, and wavelet | 2ASK, 4ASK, 8AASK, 16ASK, 2FSK, 4FSK, 8FSK, 2PSK, 4PSK, 8PSK, 16PSK, 16QAM, 32QAM, 64QAM, ASKPSK4, ASKPSAK16 | 0 | 96 | • Incorporates large set of modulation schemes • High performance with low SNR | • Uses many features • Computationally complex |
| [28] 2014 | SVM | Spectral features | 2ASK, 4ASK, 2FSK, 4FSK, 2PSK, 4PSK | 0 | 81 | • Reduced complexity | • PCC increases when SNR is increased |
| [34] 2017 | Rough set with SVM | Instantaneous, HOC, cyclostationarity, wavelet | 2ASK, 4ASK, 8ASK, 2FSK, 4FSSK, 8FSK, 2PSK, 4PSK, 8PSK, and 16QAM | 5~20 | 95 | • Enhanced performance | • Large number of samples are required. • Worse performance when the SNR is 0 dB |
| [93] 2011 | ANN | DWT | 2ASK, 4ASK, 8ASK, 2FSK, 4FSK, 8ASK, 2PSK, 4PSK, 8PSK, 4QAM, 16QAM | | 95.7 | • Reduced complexity | • Shows good performance at high SNR |
| [57] 2013 | ABC+ANN | HOC | 2PSK, 4PSK, 8PSK, 16BPSK, 4QAM, 16QAM, 64QAM | 0 20 | 57.15 76.87 | • Classifier enhancement in multipath fading • Only two features are used | • Shows good results at high SNR |
| [43] 2014 | ANN | Instantaneous features | 2ASK, 4ASK, 2PSK, 4PSK, 2FSK, 4FSK | 0 | 98 | • Enhanced performance at low SNR | • Only uses low-order modulation schemes |
| [76] 2016 | ANN | Cyclic frequency | 2ASK, 2FSK, 4FSK, 8FSK, BPSK, QPSK, MSK | 0 | 95 | • Improved performance | • Performs well at low SNR but has high computational complexity |

**Table 4.** *Cont.*

| Reference and Year | Classifier Types | Key Features | Modulation Set | SNR (dB) | PCC % | Advantages | Disadvantages |
|---|---|---|---|---|---|---|---|
| [72] 2016 | PCA+ANN | Mean value, variance, and central moments. Up to five CWTs | 4ASK, 8ASK, 16ASK, 2PSK, 4PSk, 8PSK, 16PSK, 4FSK, 8FSK, 16FSK, 8QAM, 16QAMMSK, OOK | 20 | 100 | • Reduced computational complexity | • Large set of modulations are considered. Performs well at high SNR with utilization of many features |
| [94] 2017 | PCA+ANN | Cyclic frequency | 2ASK, 2FSK, 4FSK, 8FSK, BPSK, QPSK, MSK | 0 | 95 | • Reduced computational complexity | • Increased SNR increases the recognition rate<br>• Use low-order modulation schemes |
| [35] 2015 | DT | Instantaneous; amplitude, phase, and frequency | 2ASK, 4ASK, 8ASK, 2FSK, 4FSK, 8FSK, 2PSK, 4PSK, 8PSK | 10 | 100 | • Enhanced accuracy | • Large number of samples are used. |
| [52] 2006 | DT | Fourth-order cumulant | BPSK, QPSK, 8PSK, and π/4 DQPSK | 15 | 100 | • Good results in the presence of frequency offset | • Used only for M-PSK<br>• Recognition rate will be decreased with low SNR |
| [53] 2016 | DT | Fourth-order zero-conjugate cumulant | QPSK, OQPSK, 8-PSK, and 16-PSK | 10 | 100 | • Good results in the presence of carrier and phase estimation errors. | • Used only for M-PSK<br>• Degraded performance at low SNR with small sample size. |
| [58] 2007 | DT | HOCs | 4ASK, BPSK, QPSK, OQPSK, 8PSK, 16PSK, 8QAM, 16QAM, 64QAM | 15 | 96 | • Good results in presence of carrier frequency offset | • Requires high SNR |
| [88] 2012 | DT | Fourth-order and sixth-order cumulants | BPSK, QPSK, OQPSK, 8PSK, π/4DQPSK, 16APK, 16QAM,64QAM | >10 | 90 | • Reduced computational complexity | • Requires high SNR where misclassification is shown |
| [59] 2016 | DT | Instantaneous amplitude, HOC | 2ASK, 4ASK, 8ASK, BPSK, QPSK, 8PSK | 10 | 96.6 | • Improved performance | • Degraded performance at low SNR where recognition rate drastically reduces for 4ASK and 8ASK at 0 dB SNR |
| [89] 2014 | DT | CWT | 2FSK, 4FSK, 8FSK, BPSK, QPSK, 2ASK, 4ASK, 8ASK, 16QAM, 64QAM | >2 | 90 | • Large set of modulations are considered and shows good recognition at low SNR | • Computationally complex |

**Table 4.** *Cont.*

| Reference and Year | Classifier Types | Key Features | Modulation Set | SNR (dB) | PCC % | Advantages | Disadvantages |
|---|---|---|---|---|---|---|---|
| [71] 2008 | DT | Histogram peaks in WT magnitude and mean and variance of normalized histogram | BPSK, QPSK, 8PSK, 16PSK, 2QAM, 4QAM, 8QAM, 16QAM, GMSK, MFSK | 5 | 96.8 | • Good recognition rate at low SNR | • Computationally complex and cannot be used in real time |
| [74] 2016 | DT | Wavelet variation coefficient difference and similarity feature | 2FSK, 4FSK, 8FSK, 2ASK, 4AASK, 2PSK, 4PSK, 8PSK, 16QAM | >2 | 92.39 | • Reduced complexity | • Modulations considered are easy to classify. High-order modulation schemes are not used. |
| [63] 2015 | DT | Instantaneous amplitude, kurtosis, sum-FFT | 2ASK, 4ASK, 2FSK, 4FSK, 2PSK, 8PSK | 4 | 98.8 | • Reduced computation | • Degraded performance at SNR less than 4 dB |
| [103] 2012 | GP+KNN | HOC | BPSK, QPSK, 16QAM, and 64QAM | 10 | 98 | • Good performance | • Discrimination of 16QAM and 64QAM is unsuccessful, especially under noisy conditions |
| [54] 2019 | GP+KNN | HOC | BPSK, QPSK, QAM, 16QAM, and 64QAM | 0 | 99.4 | • Enhanced performance • Reduced complexity | • Degraded performance at the lower SNR regime when the sample size is small. |
| [90] 2014 | KNN | Cyclostationarity | BPSK, QPSK, FSK, MSK | 10 | 100 | • Reduced computational complexity | • Shows high recognition rate at high SNR |
| [84] 2017 | Clustering | Constellation | 4QAM, 16QAM, 32QAM, 64QAM, 128QAM, 256QAM. | >15 | 100 | • Enhanced recognition at higher-order QAM | • Requires high SNR |
| [85] 2013 | Clustering | Constellation | 16QAM, 32QAM, 64QAM | >15 | 100 | • Enhanced performance | • Requires high SNR and cannot recognise high-order QAM |
| [106] 2009 | Clustering | Constellation | 4QAM, 16QAM, 32QAM, 64QAM | 5 | 100 | • Enhanced performance | • Cannot recognise high-order QAM |

## 6. Conclusions

This study provides a comprehensive survey of AMC techniques primarily based on FB approaches. Although likelihood techniques yield satisfactory performance, their computational complexity makes them impractical. In addition, likelihood techniques have difficulty determining the appropriate analytical solution for decision functions, especially in the case of large unknown parameters. For instance, ALRT requires multidimensional integration and GLRT requires multidimensional maximization. Thus, a large number of unknown quantities and the prerequisite of known PDFs make ALRT unsuitable for practical implementation. Maximization of unknown data may lead to the same LF values and false classification.

The FB approach refers to when relevant features are extracted and then fed to the classifier for classification. This study investigates various features used in distinguishing various modulation types and indicates their usage and applicability for specific modulation types. Instantaneous features have small computational complexity but are sensitive to noise. Therefore, these features should be combined with other features to improve their performance for identifying the signals. WT is considered to be a better approach than instantaneous features in terms of noise sensitivity, although it remains complex. HOS features are considered to be noise insensitive and less complex, especially in identifying M-QAM and M-PSK. Cyclostationary features are considered to be complex and difficult to perform in real time. Constellation shape-based features are reflected to be sensitive to noise and effective at high SNR signals.

This paper discusses the classifiers used for MT in detail, in addition to feature extraction. DTs, ANNs, SVMs, KNNs, clustering algorithms, and hybrid algorithms based on these classifiers are presented. DTs and kNNs are considered to be simple classifiers, and ANNs and SVMs have improved classification performance and are more robust to noise, but are computationally complex.

**Author Contributions:** Conceptualization, D.H.A.-N., N.A.M.I. and L.B.S.; resources, D.H.A.-N., I.A.H. and I.S.Z.A.; writing—original draft preparation, D.H.A.-N., N.A.M.I. and L.B.S.; writing—review and editing, D.H.A.-N., N.A.M.I. and L.B.S.; supervision, N.A.M.I. and I.S.Z.A.

**Conflicts of Interest:** The authors declare no conflict of interest.

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
