# Peer review of "Performance of Feature-Based Techniques for Automatic Digital Modulation Recognition and Classification—A Review"

_electronics, doi:10.3390/electronics8121407_

Round 1
Reviewer 1 Report
The paper "Performance of Feature-based Techniques for Automatic Digital Modulation Recognition and Classification – A Review" is a review of the main techniques for automatic modulation classification. The contribution of the paper is rather limited but the paper is generally well-written. The aspect of the figures should be improved.
Author Response
Dear Sir
Great Thanks and appreciation for your valuable notes stated in the reviewing.
We made some modifications and additions that were required by other reviewers and also the language of the paper was improved.
As for the contribution issue mentioned by you, we like to clarify that this is a review paper which is intended to help researchers and engineers to understand the basic concepts of the feature-based AMR and we included an extensive and detailed comparison between various techniques used in this area.
We highly appreciate any further suggestions and notes from you to make the paper even better
regards
Reviewer 2 Report
The paper can be considered for publication, given that the following comments are successfully addressed.
1) Remove Figure 2. No need for that.
2) Discuss the complexity of different methods. Include a comparison in results.
3) Include modulation classification for optical signals (e.g., the work of Prof. Lau's group in Hong Hong and the work of Xian Lin). The algorithms use features, e.g., derived from signal amplitude, along with NN, SVM, etc. for decision-making. Experimental results are also presented.
4) Additionally, the use/application of statistical tests for modulation classification, e.g., KS test, needs to be included. These are used with various features. Please, see the 2 works of Fanggang Wang (TComm & SPL) and the one of M. Mohammadkarimi.
5) English needs to be further improved/polished.
Author Response
Dear Sir
Great Thanks and appreciation for your valuable notes stated in the reviewing.
All the notes made by you were considered in the paper:
response 1
Figure 2 was omitted
response 2
A discussion for the complexity of different method was added in 4.6
response 3
Additional studies about optical signals classifications were added in section 4.3 and 4.2
response 4
Additional studies related to the concept of statistical tests were added in section 1
response 4
Improvements in the language of the paper were also made
All these changes are indicated in the new version of the paper as comment boxes
We highly appreciate all these notes that will certainly improve the quality of the paper contents and will make it more extensive.
Reviewer 3 Report
Now a lot of different type of signals are used for common communication, e.g. 2ASK, 4ASK, 2FSK, 4FSK, 8PSK, 16QAM and so on. Also a cognitive radio could be used. For receiving such signals could be used one receiver. But there exist one problem: how to recognise a kind of modulation. Automatic modulation classification (AMC) is a key procedure for present and next-generation communication networks and facilitates the demodulation process at the receiver side. However we can observe problem connected with the noises of communication channels. Two main methods, namely, maximum likelihood functions and signal statistical feature-based approach, are used for the automatic classification of modulated signals.
Authors carried out a comprehensive survey of various modulation techniques based on FB approach is conducted. They investigated a number of basic features that are usually used in determining and discriminating modulation types. Also the classifier that was used in the discrimination process is studied in detail and compared to other types of classifiers.
Analysis of obtained results shows diversified levels of immunity to disturbances for tested signals.
The review is very interesting and it can be helpful for engineers. It provides information about used modulations in wireless communication and their distinctive features which are very important from viewpoint of immunity to channel disturbances.
The references are very rich.
Author Response
Great Thanks and appreciation for your reviewing of the paper.
We made some modifications and additions that were required by other reviewers and also the language of the paper was improved.
any further suggestions and notes from you are appreciated and will be considered
regards
Round 2
Reviewer 2 Report
The paper has been improved, and it can be considered for publication.
The authors are asked to polish the language, eventually to ask the help of a colleague whose first language is English. Also, check the reference style; this needs to respect the style of the journal and be used consistently.